# Smart Greenhouse Based on ANN and IOT

**Medhat A. Tawfeek** [1,2], **Saad Alanazi** [1] **and A. A. Abd El-Aziz** [3,4,*]

1   Department of Computer Science, College of Computer and Information Sciences, Jouf University, Sakaka 72315, Saudi Arabia
2   Department of Computer Science, Faculty of Computers and Information, Menoufia University, Menoufia 12681, Egypt
3   Department of Information Systems, College of Computer and Information Sciences, Jouf University, Sakaka 72315, Saudi Arabia
4   Department of Information Systems and Technology, Faculty of Graduate Studies for Statistical Research, Cairo University, Giza 12556, Egypt
*   Correspondence: aaeldamarany@ju.edu.sa

**Abstract:** The effective exploitation of smart technology in applications helps farmers make better decisions without increasing costs. Agricultural Research Centers (ARCs) are continually updating and producing new datasets from applied research, so the smart model should dynamically address all surrounding agricultural variables and improve its expertise from all available datasets. This research concentrates on sustainable agriculture using Adaptive Particle Swarm Optimization (PSO) and Artificial Neural Networks (ANNs). Therefore, if a new related dataset is created, this new incoming dataset is merged with the existing dataset. The proposed PSO then bypasses the summarization of the dataset. It deletes the least essential and speculative records and keeps the records that are the most influential in the classification process. The summarized dataset is interposed in the training process without re-establishing the system again by modifying the classical ANN. The proposed ANN comprises an adaptive input layer and an adaptive output layer to handle the process of continuously updating the datasets. A comparative study between the proposed adaptive PSO-ANN and other known and used methods on different datasets has been applied. The results prove the quality of the proposed Adaptive PSO-ANN from various standard measurements. The proposed PSO-ANN achieved an accuracy of 94.8%, precision of 91.15%, recall of 97.93%, and F1-score of 94.42%. The smart olive cultivation case study is accomplished with the proposed adaptive PSO-ANN and technological tools from the Internet of Things (IoT). The advanced tools from IoT technology are established and analyzed to control all the required procedures of olive cultivation. This case study addresses the necessary fertilizers and irrigation water to adapt to the changes in climate. Empirical results show that smart olive cultivation using the proposed adaptive PSO-ANN and IoT has high quality and efficiency. The quality and efficiency are measured by diversified metrics such as crop production and consumed water, which confirm the success of the proposed smart olive agriculture method.

**Keywords:** dataset summarization; Internet of things; particle swarm optimization; smart agriculture; artificial neural network

## 1. Introduction

There are critical issues in the trend of greenhouse use: the continuous increase in population, decrease in agricultural land area, climate change, and water shortage. These issues negatively affect the national security of all nations. Without agriculture and the care it requires, people have no value and will perish eventually. Therefore, the trend is not only related to greenhouse use but also is related to smart agriculture. A smart greenhouse is an approach that helps guide the actions needed to transform and reorient agricultural systems in order to effectively support their development and to ensure food security in a changing climate [1]. The approach aims to address the following three main objectives: a sustainable increase in agricultural productivity, building resilience to climate change, and reducing

and eliminating greenhouse emissions wherever possible [2]. A smart greenhouse is one of the most significant areas for mobilizing resources and providing crops within the strategic goals of the World Food Organization. It aligns with the FAO's vision of food and supports the organization's plan to make agriculture more productive and sustainable [3].

This paper contributes to creating smart greenhouses using PSO, ANN, and IoT. IoT is a modern term referring to the new Internet generation that allows communication and understanding between interconnected devices [4]. These devices include sensors, advanced tools, actuators, and various artificial intelligence techniques. One of the critical features of the IoT is that it allows individuals to be free from the restrictions of having to be in the location of the devices. The person can control the tools and anything else without the need to be in a specific area to deal with special equipment [5]. The ANN is a version that simulates the communication method of biological neurons within the human brain [6]. It is a connected group of artificial cells that use a computational model to process information based on the communication method of a mathematical model of computation [7].

PSO raises problem-solving efficiency by repeatedly trying to improve a solution concerning a particular quality measure. It solves a problem by using a set of initially randomly scattered particles that move in the search depending on the mathematical procedure. Each particle is directed towards the suitable positions in the search space, which is updated when better positions are found by the particles used to solve the problem [8]. On the other hand, the existing datasets used in the cultivation process are massive, and new species are constantly emerging, which needs to be collected and summarized. If there is a new incoming training dataset, it will be processed first as follows. The representation of incoming datasets is unified by removing useless and unused records and merging these datasets with legacy datasets. Then, the summarization process takes place. This summarization process is challenging when using traditional algorithms, but it will be easier using modern algorithms.

In this paper, an adaptive PSO-ANN for smart agriculture is proposed. The proposed combined approach is used for dataset summarization and classification processes. The proposed adaptive PSO-ANN is implemented in various datasets and compared with other known classification methods. The experiments provide evidence of the proposed adaptive PSO-ANN's quality from multiple measurements, such as classification accuracy, precision rate, recall rate, F1 score, and absolute accuracy error. The smart olive cultivation case study by the proposed adaptive PSO-ANN is introduced. The performance evaluations prove the efficiency of the olive cultivation by the proposed method and IoT technology from diversified metrics, such as the amount of olive production by weight and the size of olives and the consumed irrigation water. The layout structure for the rest of this research is as follows. Section 2 reviews the scientific backgrounds that overview the olive tree, smart agriculture, PSO, ANN, and IoT disciplines. The literature review is presented in Section 3. The contribution of summarizing datasets by the proposed adaptive PSO-ANN that are used in smart agriculture is proposed in Section 4. Section 5 investigates the performance evaluations of the experiments applied to various datasets. Section 6 introduces the smart olive cultivation case study by the proposed Adaptive PSO-ANN. The conclusion of this research paper and future work orientations are tackled in Section 7.

## 2. Scientific Background

In this Background section, five related topics, namely the olive tree, smart greenhouse, PSO, ANN, and IoT, are presented in separate subsections.

### 2.1. Olive Trees and Olive Fruit

The tree is truly blessed for having many benefits found in its oil, wood, and leaves. Olives are excellent for eating after being washed and pickled. Olive oil is one of the most luxurious oils. It is harmless; on the contrary, it benefits the heart and all body parts [9].

Integrating the soil and maintaining healthy, fertile, well-structured soil are starting points in preventing pests and diseases threatening olive trees. A farmer must focus on nourishing the soil by treating it as a living medium. It is essential to avoid the excessive use of chemical fertilizers. Although plants that feed on nitrogen in their chemical form overgrow, their cell walls become thin and weak. Fertilization with excessive chemical nitrogen leads to an imbalance in proteins and carbohydrates in plants. Adding abundant chemical fertilizers to olive trees increases their vulnerability to drought as water needs increase. It kills the beneficial microorganisms in the soil and loses the organic matter that fertilizes the soil and maintains its moisture. This leads to a decline in productivity and the emergence of the olive branch weevil and the olive fruit fly. Chemical fertilizers are just as dangerous to public health and the environment as chemical pesticides [10].

Therefore, although nitrogenous chemical fertilizers (nitrates and nitrites compounds) are considered one of the most important fertilizers, they pollute the environment and groundwater with nitrates [10]. Based on the foregoing information, it is self-evident that a focus on agriculture in general and on olive cultivation is needed, with great care in the use of chemical fertilizers.

Smart Agriculture

Innovative technologies play crucial roles in helping to meet the growing nutritional needs of the world's population, and these involve data management, analysis systems, and remote-control technologies. Moreover, with the use of the most prominent technologies of the Fourth Industrial Revolution, such as artificial intelligence, robotics, and IoT, agriculture has become more productive, profitable, less harmful to the environment, and consumes fewer land resources [11]. Smart agriculture can be defined as a system that relies on advanced technology to grow food in clean ways. It directs the use of natural resources, especially water and fertilizers. Its reliance on management and information analysis systems to make the best possible production decisions at the lowest costs is one of its most prominent features, and it helps in automating agricultural processes such as irrigation, pest control, soil monitoring, and crop monitoring [12]. Smart farms have real potential in delivering sustainable agricultural production based on a more resource-efficient approach [2]. The Arab region is one of the regions that face major environmental crises, such as a lack of arable water, climate change, drought, and desertification. This negatively affects the provision of food and the achievement of food security. Therefore, it can be said that the Arab region is the region that needs to apply smart farming techniques the most. The financial cost is one of the difficulties encountered, which constitutes a handicap for many nations. The success of smart agriculture requires the exchange of knowledge, learning more ideas about IoT, and the inclusion of communication and information technology as fundamental factors for the development of agriculture [2]. Smart agriculture will witness increased growth in the coming years. It is similar to all modern smart technologies that seek to develop and prosper in society, and it is needed secure to secure all basic needs. The spread of modern technologies, their access to various groups, and their ease of use will contribute to the adoption of smart agricultural practices. Consequently, this will bridge the food gap caused by an increase in the population and the scarcity of resources. Smart agriculture has had immense success with IoT technologies as it has devices that monitor temperatures, lighting levels, humidity, and the pressure of air and water consumption [4]. A farmer or farm owner can remotely control appliances to raise or lower the temperature, control the level of lighting, and open and close windows via the Internet. Smart agriculture will lead to an abundance in resources during green revolutions and save humanity from the consequences of population explosion and environmental degradation.

## 2.2. Internet of Things (IoT)

IoT is one of the leading technologies used in smart farming. It is the process of connecting any device to another device over the Internet. These devices range from cell phones to home appliances and machinery used in factories and agricultural fields.

They can be operated and controlled as data can be sent and received from them over the Internet [13]. One of the most prominent applications of IoT is smart agriculture. It is used in farm management and crop control via information and communication technology, sensors, remote control systems, and autonomous machines. The goal is to obtain accurate data and to invest these data in accurately directing agriculture towards greater production with lower costs and higher quality. For example, remote sensors placed in fields allow farmers to obtain detailed maps of both the terrain and resources in an area. It can also measure variables, such as soil temperature and humidity, and it can predict weather patterns for the coming days and weeks [14]. IoT-based agriculture helps make better decisions in improving agricultural production. In addition, the collected data and the analyzed data play a leading role in monitoring farming pests and determining the exact number of pesticides required to avoid overuse. Data collection and analysis processes can also be rationally applied to irrigation water. Everything falls under the IoT concept, such as clothes, furniture, household items, body parts, streets, and even animals. Anything that can stick to a processing unit and Internet connection feature is a thing in the world of the IoT [15]. The IoT is the concept of how we live and manage our business using the Internet. A combined IoT system includes five main components, which are stated as follows:

- Physical devices (sensors, actuators, etc.);
- Connection;
- Data processing;
- Statistical processes;
- Interface.

As for how the IoT works, it starts with sensors assembling data from the environment in which they are located. These data are sent across the network to the unit that provides numerous services to individuals or companies where the sensors are connected to the network [4]. The connection can be made with several methods, including smartphones, satellites, Wi-Fi, or Bluetooth. Once the data reach the responsible processing unit in the network, it is processed. The process of processing these data may be simple or complex depending on the amount and type of data obtained. After that, some needed statistics are plotted. Finally, the results are sent to the end user in the form of a specific alert so that the user can change or modify the settings of the devices, and sometimes these devices are limited automatically without the need for human intervention [5]. In the future, the IoT will control the daily operations of our lives, as well as the processes of the industry, agriculture, etc., and it will increase their efficiency. There will be a significant reduction in many important factors, such as less time lost due to unexpected system or machine breakdowns, energy consumption, and the effort made to move to the workplace. The real-time access and collection of data lead to making important business decisions remotely [11].

*2.3. Artificial Neural Network(ANN)*

An ANN is similar to human brain behavior in that it acquires knowledge by training and stores this knowledge using connecting forces within neurons called synaptic weights. There is also a vital neuronal similarity, providing biologists with the opportunity to rely on ANN to understand the evolution of biological phenomena [16]. Just as humans have senses that act as input units that connect them to the outside world, neural networks need input units and processing units. The input units create a layer called the input layer, and the processing units form the processing layer, which produces the output of the network. Between each of these layers, there is a layer of interconnections that links each layer with the next layer [7]. In the interconnections layer, the weights of each interface are set. The network contains only one layer of input units, but it may contain more than one layer of processing layers. In the processing layers, calculations are performed by adjusting weights, and by using them, we obtain the appropriate reaction for each instance of the input in the network [17]. The weights represent the basic information that the network will learn. Weights need to be updated during the training phase, and many different

algorithms are used for this update process depending on the type of network [18]. One of the most important of these algorithms is the back propagation algorithm, which is used in training fully interconnected, forward-fed, multi-layered, and nonlinear neural networks. This algorithm is considered as a generalization of the error correction pattern-training method. Moreover, this algorithm is implemented through two main stages:

- Feed forward propagation;
- Back propagation stage.

In the forward deployment stage, there are no adjustments made relative to the network's weights. This stage begins with feeding the network through the input layer and then processing them using the current weights, followed by forward diffusion to the rest of the network layers. The back propagation stage is the stage of adjusting the weights of the network. The role of back propagation is due to how the deflection is calculated for the multiple network layers. The signal is re-propagated from the output to the input in reverse, during which the network's weights are adjusted [19]. Before training the network, initial weights must be set automatically. After determining the initial values of the weights, the network becomes ready for training. During training, these weights repeatedly change until reaching the smallest value of the Mean Square Error (MSE) or reaching one of the stopping criteria used during training [18].

### 2.4. Particle Swarm Optimization (PSO)

Swarm particles hover in the environment to follow experienced swarm members, and their movement is directed toward good feeding areas in their environment. The PSO algorithm simulates this idea and aims to exchange information and share experiences about a search to seek a place that represents an efficient solution. This is realized by designating a rudimentary random position in the search space and impetuous velocities for every particle. Figure 1 shows particles with their related positions and velocities.

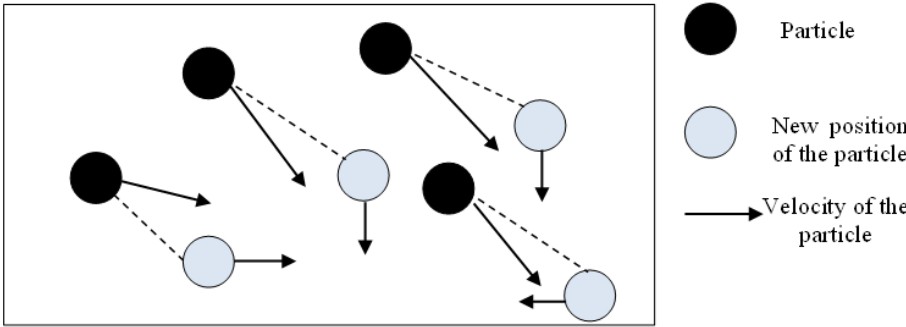

**Figure 1.** Particles with their related positions and velocities.

It is determined that the loop iteration of the PSO algorithm involves moving the particle around the decision space. The Pseudo code of PSO is shown in Figure 2.

```
Initialize all the particles randomly.
Repeat
    Evaluate the fitness function f(x)
    For each particle
        Update velocities by Eq. (1)
        Use the updated velocities to update the positions of
particles.
        If f (x_i) < f (P_Best_i) Then update P_Best_i
        If f (x_i) < f (G_Best) Then update G_Best
    End For
Until stopping criteria
```

**Figure 2.** Pseudo code of PSO.

The algorithm follows a global best value to indicate which particle data are closest to the target and the cutoff value to indicate when the algorithm ends. Depending on how far the particle's position is from the target, a velocity value is deliberated by Equation (1) in the standard PSO algorithm [8]:

$$V = V(current) + U_1 C_1 x (P\_Best_i - X_i) + U_2 C_2 x (G\_Best - X_i) \tag{1}$$

where $V$ stands for the calculated velocity of the particle, and $V(current)$ holds the current velocity of the particle. $U_1$ and $U_2$ are two random variables that hold values in the range from zero to one. $C_1$ and $C_2$ are two constant learning factors. $X_i$ keeps the particle's current position within the search area. $P\_Best_i$ is the best value realized by the particle. G_Best is the global best value realized by any particle [20].

## 3. Literature Review

Information Gain (IG) was introduced in the mid-fifties, and the main idea was to take the measurements of the amount of information gained about a reckless variable from the oversight of another reckless variable [21]. Principal component analysis is a statistical procedure used to reduce the dimensions of a dataset consisting of many variables that correlate with each other, either significantly or slightly, while retaining, to the greatest extent, the variance present in the dataset [21]. Nature-inspired algorithms, as they are known by their names, are approaches driven by the properties of nature's directives. They are used to reach near-optimal solutions to hard problems using methods developed in nature. These algorithms can be classified into two classes: evolutionary and swarm-based [22]. The genetic algorithm (GA) is the most common type of the first class. It imitates natural selection to create a next-generation [23]. The ant colony optimization (ACO) in [24], particle swarm optimization (PSO) in [20], and artificial bee colony (ABC) in [25] are the most common algorithms in the second class. Supportive Vector Machine (SVM) is the most popular and powerful machine learning algorithm. This method is based on a sound theoretical foundation that is persistent in statistical learning theory. SVM is not sensitive to the number of dimensions [8].

The authors in [26] proposed learning a prediction model based on ANN for improving the precision of prediction techniques in powerful conditions. The proposed model is made of a prediction module and a learning module. The learning module is mindful of consistently inspecting the prediction module. It tunes its exhibition by evaluating its results along with whatever is on the outside boundaries that can influence its exhibition. Three deep neural network architectures have been adapted in [16] for energy classification. They are LSTM, denoising autoencoders, and a network. Seven metrics were used to test the performance of these algorithms on the true total power data from five instruments. The methodology for identifying residential equipment was introduced in [18]. This proposed method is used for monitoring systems. The proposed system can perform the task of identification and provides satisfactory results. ANN is used for disease classification,

as in [6]. It uses an omics dataset and states that MLPs with ample hidden neurons are efficient in increasing classification performances for the purpose of diagnosis. ANNs are used in [17] to enhance the prediction of intensive care deaths. This approach outperforms the SAPS 3 model. It promotes the dependence of intensive care registries on ANN models for short death predictions. In [27], a smart device is set in the middle plants' path so that it will cover the entire ranch in a crisscross manner. In this proposed system, Raspberry Pi is a single-board PC utilized in this framework associated with the Arduino microcontroller. Raspberry Pi is associated with a camera module, Wi-Fi, an image processing unit, and a memory card. While Arduino is associated with a DC engine, Spraying Trigger, and sensors, two sensors are predominantly utilized: for example, a weight sensor and a distance sensor. The smart sensor incorporates this camera and a vermin sprayer, which will travel through this link. It catches any abnormality in the plant and advises the rancher through an Android application. On the off chance that any unfamiliar material is discovered, appropriate posting on the plant is finished utilizing a sprayer equipped in the device on the rancher's endorsement. The proposed approach is dependent on IoT and makes the refreshes accessible on the web via an Android application. In this paper, a smart olive greenhouse using ANN and IoT is implemented as a real project. The research in [28] provides a review of the current majesty of IoT for on-farm mensuration, the conditions of the successful enforcement of IoT in countries that have low or middle income, difficulties related to IoT implementation in small farms, and recommendations for practitioners. The authors of [28] concentrate on the functions of smart technology that can be applied in sustainable agriculture. They use a case study named "Wangree Health Factory Company." This case study circulates a plant equipped with artificial lighting technology to grow fresh organic vegetables and fruits. The authors evolve an analytical framework for analyzing how plant deployment affects the sustainability of agriculture.

## 4. Smart Agriculture Based on Adaptive PSO-ANN

If there is a new incoming training dataset, it is first pre-processed as follows:
- The representation of the incoming datasets is unified by removing useless and unused fields;
- Merging this dataset with the old datasets.

After that, the dataset summarization is performed by the proposed PSO, as shown in Figure 3. The representation of particles is as follows. The length of the particle is equal to the number of records since the particle is considered to be binary; that is, each cell in the particle is either one or zero. A value of one indicates that a record will be present in the dataset, and zero indicates that this record will not exist.

```
Input: the merged dataset
Output: summarized dataset
The initial part:-
    (1) Remove the duplicated records in the merged dataset.
    (2) Remove conflicting records
    (3) Generate a random solution for each particle
    (4) Set value for Ps, Tmax, and Vmax
    (5) Set value for RTs
    (5) Set iteration=1
    (6) Set P_Best=0.
    (7) Set G_Best=0.

The iterative part:-
    While (iteration <= Tmax) do
    Begin
            (1) For i=0 to i= Ps do
                Measure the fitness value of the particle according to RTs
                If the computed fitness value > P_Best
                    Set P_Best = computed fitness value
                End If
                If the computed fitness value > G_Best
                    Set G_Best = computed fitness value
                End If
                End For
            (2) Sort particles in descending order by their p_Best scores
            (3) For i=0 to i= Ps do
                Compute the velocity of the particle by Eq. ( 2)
                Update the particle depending on the computed velocity
                End For
            (4) Increment iteration by one
    End While
    Output the dataset according to the G_Best
```

**Figure 3.** Proposed PSO algorithm for dataset summarization.

It is clear from the proposed PSO that *P_Best* is gradually approaching the *G_Best* value. *G_Best* gradually reaches the target when the algorithm reaches *T_max*, which represents the number of iterations and works as a stopping criterion. Ps represents the number of particles. A different shape for how to compute velocity is used in the proposed algorithm. The velocity score is calculated using *Best_fitness*, *V_max*, and *P_Best*, as shown in Equation (2):

$$Velocity = \frac{V\_MaxXP\_Best}{Best\_fitness} \tag{2}$$

where *Velocity* is the calculated particle's velocity, *P_Best* is the fitness value of the particle. *Best_Fitness* is the best particle's fitness value that is equal to the *G_Best* value. The proposed function for updating the velocity specifies it as a measure of how poorly each particle is performing and how it is moving faster than the best-performing particle. It is shown in Equation (2) that velocity is determined as a measure of how well each particle is performing. Each particle moves according to its own performance.

The accuracy classification of ANN with a sample of records as a test dataset (RTs) is used to evaluate each particle's fitness. The records for tests can be selected randomly for the entire dataset, but in this research, the records with high ranks in the dataset are selected. A column called "ranking" has been added to the dataset to help identify which records have high ranks. The rank is incremented if this record appears in the incoming dataset. RTs determine the number of records used in the test for determining classification accuracy. The proposed adaptive ANN is worked as a learning tool. It learns from the current as well as future experiences with the proposed PSO to make the right decision. The proposed ANN can be retrained if new datasets emerge while it is already working on making various decisions. Initially, the proposed ANN is trained with the available data. Training is performed via a smart device, such as a computer or a smartphone. After completing the training and determining the internal weights of the ANN, the running mode starts,

where the decision is made based on the readings from the different sensors and the current weights of the network. When the new dataset appears, pre-processing and dataset summarization take place. After that, adaptive ANN training begins again. Figure 4 shows the proposed adaptive ANN structure. The size of the input and the size of the output layer are adaptive, which means that they can be varied depending on the available dataset. The proposed ANN has units that are fully connected. Every unit in the input layer is connected with every unit in the first hidden layer. Each unit in the first hidden layer is connected with every unit in the second hidden layer, and so on. All units of the proposed ANN are sigmoid units. The sigmoid unit is constructed as in Figure 5. The first part of the sigmoid unit calculates a linear set of its inputs. After that, it stratifies a threshold to the computed value. The proposed ANN has been trained by the back propagation algorithm to extract the affluent diversity of nonlinear decisions.

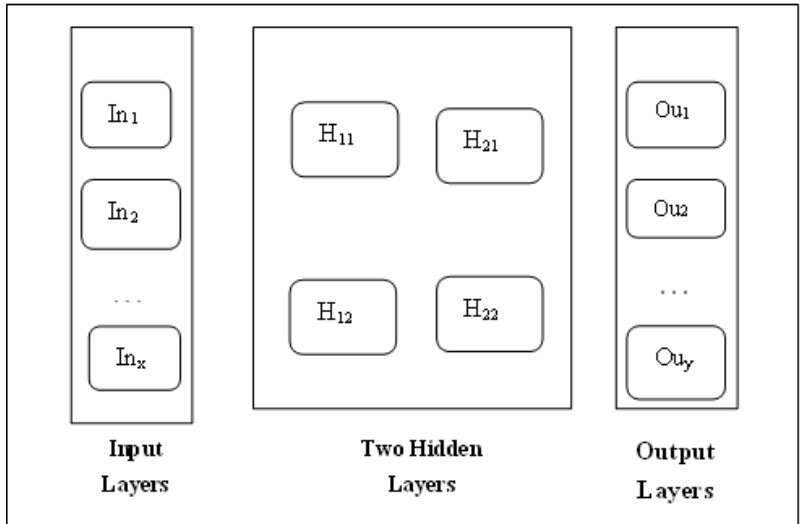

**Figure 4.** The proposed adaptive ANN structure.

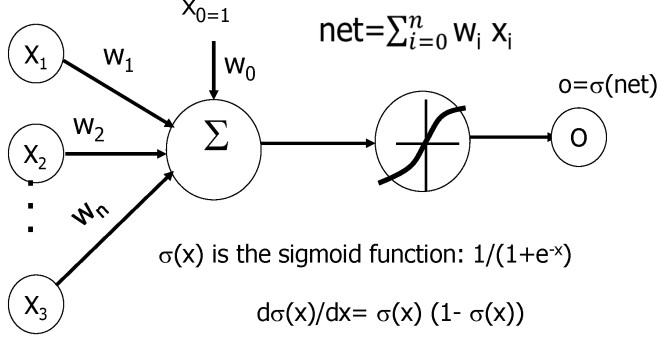

$$net = \sum_{i=0}^{n} w_i \, x_i$$

$$o = \sigma(net)$$

$\sigma(x)$ is the sigmoid function: $1/(1+e^{-x})$

$$d\sigma(x)/dx = \sigma(x)\,(1 - \sigma(x))$$

**Figure 5.** The sigmoid unit.

The reason for depending on the back propagation algorithm is that it has been proven to generate excellent results in many real-world applications. The algorithm tries to reduce the squared error between the output and the target values. The updating weight rule of

the back propagation algorithm has been modified in the proposed ANN. It depends on a second term called momentum, as shown in Equation (3):

$$\triangle w_{ij}(n) = \eta \, \sigma_j \, x_{ij} + \alpha \, \triangle w_{ij}(n-1) \tag{3}$$

where *n* refers to the loop iteration of the back propagation algorithm, $\triangle w_{ij}(n)$ is the computed value for updating the weight value, $\eta$ refers to the learning rate that has a small constant positive value, $\sigma_j$ is the error term of unit *j*, and $\alpha$ is the momentum that has a constant value between zero and one. The main effect of using a momentum term is to keep winding in the same orientation from the previous iteration to the current one. Moreover, it is used to deal with the problem of being trapped in the local minima. Another remediation is used to handle the local minima problem. Multiple versions of the proposed ANN are created. Each version is trained with the same dataset but with different initial random weights. The version that has high performance is selected in the process of classification.

## 5. Performance Evaluations

Different datasets were used to measure the performance of the proposed system, and they are as follows:

- MNIST is the most common dataset used in the machine learning process. It is a dataset containing more than fifty thousand examples for training and 10,000 examples for testing.
- NSL-KDD is an improved version of KDDCUP, and it is proposed to manipulate some of the issues in the KDDCUP dataset, such as the redundancy that negatively affects the performance of the evaluated systems.
- Syngenta Crop Challenge 2017 in [29] includes 2267 maize hybrids at 2122 locations between 2008 and 2016, along with weather and soil conditions.
- Plant datasets are related to the University of Arkansas in [30].

The following metrics are used to evaluate the performance by checking its ability to perform valid predictions. The classification accuracy (CA) is the ratio of true classified instances to the full number of instances. Equation (4) is used to measure the CA. The CA states whether the proposed model has the ability to take correct and appropriate actions depending on the current state:

$$Classification\,Accuracy = \frac{TP + TN}{TP + TN + FP + FN} \tag{4}$$

where *TP* denotes how many records are positive and are correctly classified as positive. *TN* denotes how many records are negative and are correctly classified as negative. *FP* denotes how many records are positive and are incorrectly classified as negative. *FN* denotes how many records are negative and are incorrectly classified as positive.

Precision Rate (PR), Recall, F-Measure, Absolute Accuracy Error, Mean Absolute Error, and Absolute Precision Error are considered standard performance evaluation [31]. The control parameter of the proposed PSO for dataset summarization that should be finetuned is shown in Table 1. RTs is set to 5% from the total size of the dataset, which means that 5% of high-ranked records are chosen for the test to determine the classification accuracy.

**Table 1.** Control parameters values of the PSO.

| Parameter Name | Value |
| --- | --- |
| Ps: Number of particle | 100 |
| $T_{max}$: Iterations number | 100 |
| $V_{max}$ | 5 |
| ine RTs | 5% |

The experimental results indicate that the proposed PSO-ANN can summarize more than 25% of the dataset's size while maintaining classification accuracy without affecting other measurement parameters. The accuracy of the summarizing process is affected by about 0.7%, as shown in Tables 2 and 3. This percentage of less than one percent is exceedingly small and does not affect the system. On the other hand, the size of the data is reduced, which speeds up the training process. An uncontrolled increase in the size of the dataset (i.e., redundancy, inconsistency, and unimportant records) still negatively affects the classification process. Summarizing the dataset helps solve the problem of overfitting. Entire records of a dataset do not contribute equally to training and classification procedures, as some may contribute well and effectively, and others may not contribute to anything other than its negative impact. The proposed PSO-ANN has a satisfactory performance than the Hybrid algorithm in [32], the SVM algorithm in [8], the Ensemble Learning algorithm [33], the GA algorithm [23], and the CGB algorithm in [34], as shown in Table 4. The experimental results indicate that the proposed PSO-ANN can summarize more than 25% of the dataset size while maintaining the classification accuracy without affecting other measurement parameters. After summarizing the process, the accuracy is affected by about 7%. This percentage of less than one percent is exceedingly small and does not affect the system. On the other hand, the size of the data is reduced, which speeds up the training process. An uncontrolled increase in the size of the dataset (i.e., redundancy, inconsistency, and unimportant records) still negatively affects the classification process. Summarizing the dataset helps solve the problem of overfitting.

**Table 2.** The classification performance of the ANN without dataset summarization on MNIST dataset.

| Evaluation Metrics | PSO-ANN Performance |
| --- | --- |
| Accuracy (%) | 95.59 |
| Precision Rate (%) | 94.85 |
| True Positive Rate (TPR)-Recall (%) | 97.4 |
| True Negative Rate (TNR) (%) | 93.2 |
| False Negative Rate (FNR) (%) | 0.03 |
| False Positive Rate (FPR) (%) | 0.07 |
| Mean Absolute Error (MAE) (%) | 2.67 |
| F1-score (%) | 96.11 |
| Size of dataset Summarize | 31% |

**Table 3.** Classification performance of the ANN with dataset summarization on MNIST dataset using the proposed PSO.

| Evaluation Metrics | PSO-ANN Performance |
| --- | --- |
| Accuracy (%) | 94.8 |
| Precision Rate (%) | 91.15 |
| True Positive Rate (TPR) - Recall(%) | 97.93 |
| True Negative Rate (TNR) (%) | 98.75 |
| False Negative Rate (FNR) (%) | 0.02 |
| False Positive Rate (FPR) (%) | 0.09 |
| Mean Absolute Error (MAE) (%) | 3.91 |
| F1-score(%) | 94.42 |
| Size of dataset Summarize | All dataset |

**Table 4.** Performance evaluation of the proposed PSO-ANN and other related classification methods.

| DataSet | Evaluation Metrics | PSO-ANN Proposed | Hybrid [32] | SVM [8] | Ensemble Learning [33] | GA [23] | CGB [34] |
|---|---|---|---|---|---|---|---|
| Plants Dataset | Accuracy | 93 | 86.2 | 84.9 | 87.87 | 80 | 70.87 |
| | Precision | 88.1 | 89.7 | 93 | 85.2 | 78.8 | 86.3 |
| | Recall | 80 | 77.3 | 65 | 71.1 | 0.69 | 54 |
| | F1-score | 83.8 | 83 | 76.5 | 77.5 | 73 | 66.4 |
| | MAE | 11 | 15.7 | 17.8 | 10.6 | 13.9 | 18.9 |
| KDD Dataset | Accuracy | 98.5 | 94 | 96 | 97 | 87 | 90 |
| | Precision | 92.3 | 91.2 | 93.8 | 90.8 | 90.9 | 89.9 |
| | Recall | 88.1 | 87.5 | 83.2 | 79.9 | 81 | 86.2 |
| | F1-score | 90.15 | 89.31 | 88.1 | 85 | 85.6 | 88 |
| | MAE | 16 | 17.3 | 24.5 | 20.2 | 15.9 | 26.7 |
| Syngenta Dataset | Accuracy | 82.5 | 77.1 | 79 | 69 | 72 | 79 |
| | Precision | 78.8 | 65.2 | 60 | 70 | 53 | 68 |
| | Recall | 63 | 56 | 59 | 61 | 59 | 64 |
| | F1-score | 70.02 | 60.25 | 59.50 | 65.19 | 55.84 | 65.94 |
| | MAE | 20 | 28.1 | 25.3 | 27.91 | 29 | 33.1 |

Whole records of a dataset do not contribute equally to the training and classification procedures, as some may contribute well and effectively and others may not contribute anything other than its negative impact. The proposed PSO-ANN has a satisfactory performance than the hybrid algorithm in [32], the SVM algorithm in [8], the ensemble learning algorithm [33], the GA algorithm [23], and the CGB algorithm in [34]. The efficiency of the proposed system appears with the emergence of different and new datasets related to one domain and collected into one unified dataset, which is more useful and efficient. Important factors that helped in the efficiency and success of the proposed method are as follows. The proposed PSO removes duplicate records and conflicting records and summarizes the dataset in an intelligent manner. Repeated records affect the training process by causing a deviation in adjusting the weight values during the training process. Conflicting records lead to a terrible dispersion of weight values. The proposed PSO in the dataset summarization process removes these two types of data from the dataset and influential minor records, and it keeps records that are of high importance in the training process. The proposed ANN is consumed in determining the classification accuracy in the summarization process. The sigmoid units, the back propagation, and the momentum increase the efficiency and activate the behavior of the proposed ANN for training and classification processes.

## 6. A Case Study of Smart Olive Cultivation

This case study has been applied only to the type of Picual olive tree that is also known as Marteña or Lopereña. This type was chosen because it contains high oils. Picual olive trees represent 25% of the world's total olive oil production. Figure 6 outlines the main operations in smart olive cultivation by the proposed adaptive PSO-ANN and IoT. If there is a new incoming training dataset, it is firstly pre-processed and summarized.

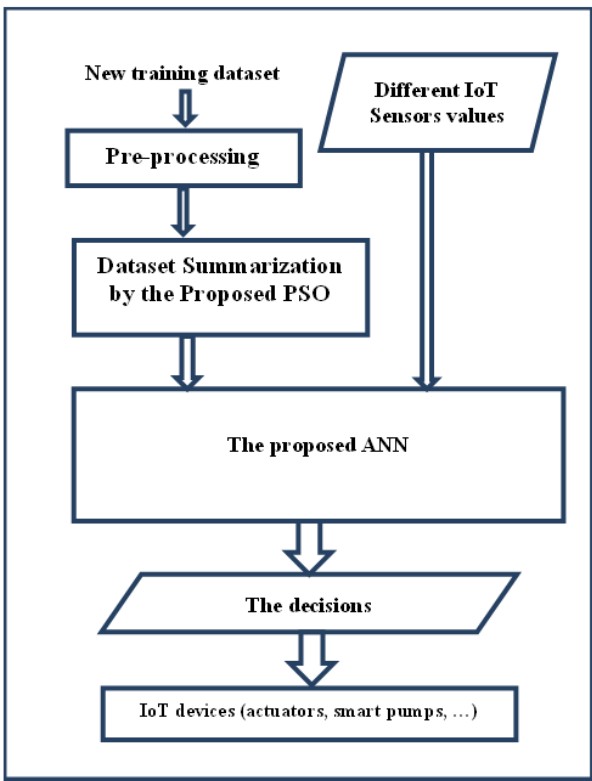

**Figure 6.** Smart olive cultivation by the proposed adaptive PSO-ANN and IoT

Two units are used in the output layer of the proposed ANN. The first unit represents the decision for irrigation. The second unit represents the decision for fertilization. Only liquid fertilizers were used. For the input layer, there are five units that represent the olive's type, temperature, soil moisture, wind, and humidity. Most modern IoT tools and devices are used to improve the entire agricultural process of olive trees. It automatically contributes to elevating the total monitoring related to the surrounding climate and environmental changes. It also helps in completing the decision resulting from the proposed ANN with respect to irrigation, fertilization, or both without human intervention. A set of dedicated sensors was used to read the current image surrounding olive trees, and it is sent to the proposed ANN to make the appropriate decision. Four types of sensors are used. The first type is LM35, which is used to measure the temperature. The second type is a hygrometer, which is used to measure the moisture of the soil. The third type is OMC-118, which is used to measure the force of the wind. The fourth type is DHT22, which is used to measure the degree of humidity. In addition, two tanks are used. The first tank is large for water. The second tank is a small size for filters. Each tank has a pump.

Two agricultural plots with the same area were used. The width of one plot area is 10.5 m, and its length is 17.5 m. The proposed smart olive cultivation by the proposed adaptive has been established in the first plot, and the manual olive cultivation method is applied in the second plot. A ready-made seedling method was used: The seedling was 12 months old, and the seedling's length was 135 cm. Fifteen ready-made seedlings of the same species, approximately the same age and height, were used for each plot. The pits were dug and prepared in two days, and the seedlings were planted in parallel at the two plots at the same time. The distance between any two seedlings was 3.5 m. The experimental results of cultivating olive trees by the proposed method and the traditional method are based on the following measures:

- Production amount of the olive fruit;
- The average weight of the fruit;
- Average longitudinal diameter and cross diameter;

- The amount of irrigation water.

The following results are the first signs of cultivated seedlings. The production amount of the olive fruit (harvest of the olive fruit) of the proposed smart olive greenhouse (SOG) using ANN and IoT and the Traditional Olive Cultivation (TOC) is shown in Figure 7.

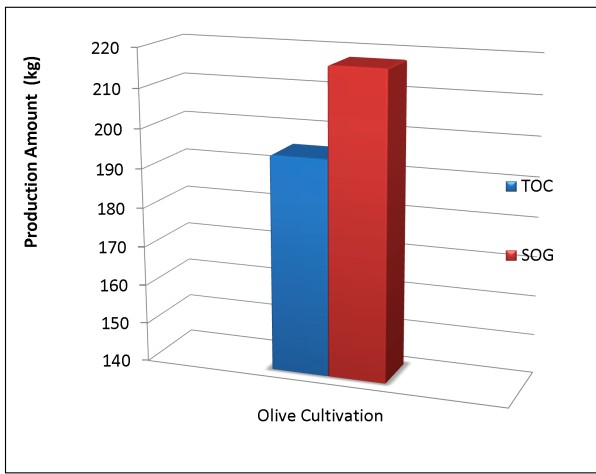

**Figure 7.** Production amount of SOG and the TOC.

The average weight of the olive fruit from the SOG and TOC is presented in Figure 8. One hundred olives were randomly assigned from SOG and TOC separately for mean olive weight measurements. They were weighed and divided by their number. This process was repeated several times, and the results were averaged and documented in Figure 8. The average longitudinal diameter and cross diameter of SOG and TOC are viewed in Figure 9. Ten olives were randomly selected to measure the average longitudinal diameter. The longitudinal diameter of each fruit was measured, and the average results were taken. This process was performed separately for SOG and TOC methods.

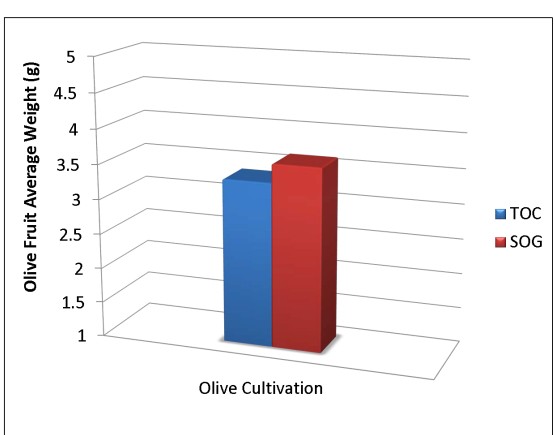

**Figure 8.** The average weight of the olive fruit of the proposed SOG and TOC

The same steps were also applied to measure the crossed diameter. An increase in longitudinal diameter or cross diameter or both together indicates that the fruit is vigorous and was cared for during the growth period.

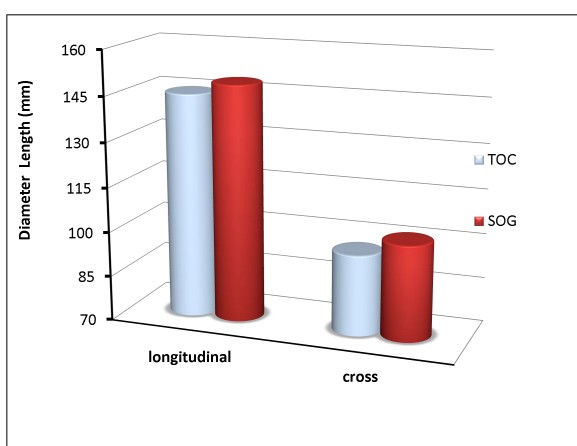

**Figure 9.** Average longitudinal diameter and cross diameter of the proposed SOG and TOC.

The amount of irrigation water needed throughout the season of SOG and TOC is shown in Figure 10. Saving the consumed amount from irrigation is important, especially in areas without rivers and suffering from water poverty; thus, more savings are better.

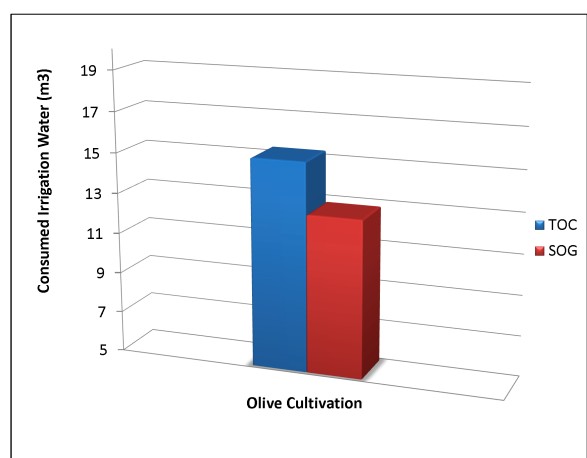

**Figure 10.** The amount of irrigation water of the proposed SOG and TOC.

The obtained results show that the proposed smart olive greenhouse (SOG) using adaptive PSO-ANN and IoT has a greater amount of olive production. The average weight of their olives is heavy. The average longitudinal and cross diameter is long. Moreover, it fully saves the amount consumed from irrigation water. The decrease in irrigation water consumption, along with the increase in olive production, proves the efficiency and robustness of the proposed SOG method. One of the most important things that helped in the efficiency and success of the proposed method is the use of modern tools from the IoT, in addition to the proposed adaptive PSO-ANN that is trained on previous experiences in the field of olive tree cultivation. The proposed SOG does not only rely on the currently available dataset but also on any other data that emerges. The proposed SOG takes the new incoming dataset into account in addition to previously acquired experiences. Taking information that is related to irrigation and fertilization as effective areas in the training dataset contributes to the prosperity of the proposed SOG using ANN and IoT. This is because of the negligence of irrigation water or the incorrect utilization of the amount of fertilizers that may damage the olive plant. The proposed SOG is also adaptive to environmental changes and can handle this issue appropriately and quickly.

## 7. Conclusions and Future Work

Smart agriculture exploits the means of modern technologies and allows it to self-respond to the challenges of climate change in specific locations, make automatic decisions, and increase productivity. This research proposed smart agriculture by exploiting the proposed adaptive PSO-ANN and IoT. The main objective of the proposed PSO is to summarize the combined datasets related to one domain in an efficient manner. It eliminates the negative impact of large-volume datasets by treating redundancy, inconsistency, and unimportant records. The main target of ANN training is to be more active and efficient in making the right decisions for the daily needed operations that are used in farming. The proposed ANN can train again as new datasets emerge while the system is fully operational on the ground. The proposed adaptive PSO-ANN is implemented and compared with some known methods in the literature in different datasets. The results of different experiments indicate that the proposed PSO-ANN can summarize more than 25% of the dataset's size without affecting the entire performance. The proposed PSO-ANN is a leading and remarkable comparative method in terms of accuracy, precision, recall rate, F1-score, and mean absolute error at 94.8%, 91.15%, 97.93%, 94.42%, and 3.91, respectively. The proposed model is implemented in smart olive cultivations. Two major olive cultivation processes were examined, irrigation and fertilization, to enhance the degree of olive growth. After dataset summarization and training, the new weights were fed to the control circuit, which also takes other inputs from various modern IoT devices. These devices, such as sensors besides tanks, pumps, and actuators, have been installed at several positions on the farm. The efficiency of the smart olive cultivation based on the proposed model has been demonstrated via various measures such as the production amount of the olive fruit, the average weight of the fruit, the average longitudinal diameter, the cross diameter, and the amount of irrigation water. In future, PSO enhancements will be considered to increase the efficiency of dataset summarization. Other methods, such as ant colony optimization and artificial bee colony, may be combined with the proposed PSO to achieve the same goal.

**Author Contributions:** Conceptualization, M.A.T. and S.A.; methodology, M.A.T.; software, M.A.T., S.A.; validation, M.A.T., A.A.A.E.-A.; formal analysis, A.A.A.E.-A.; investigation, S.A.; data curation, A.A.A.E.-A.; writing—original draft preparation, M.A.T.; writing—review and editing, A.A.A.E.-A.; visualization, A.A.A.E.-A.; supervision, A.A.A.E.-A.; project administration, M.A.T. All authors have read and agreed to the published version of the manuscript.

**Funding:** This research received no external funding.

**Data Availability Statement:** Not applicable.

**Conflicts of Interest:** The authors declare no conflict of interest.

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
