# Peer review of "Smart Greenhouse Based on ANN and IOT"

_processes, doi:10.3390/pr10112402_

Round 1

Reviewer 1 Report

The file is attached.

Reviewer 2 Report

In this research, The authors are studying the smart greenhouse based on ANN and IOT. The study in this paper mainly contributes to the process of creating smart greenhouse of olive trees using Artificial Neural Networks (ANN) and IoT. However, there were still some shortcomings:

1. How to obtain the yield of olive fruit ? Compared with the traditional cultivation methods, what are the similarities and differences of olive cultivation in smart greenhouse ?

2. Fruit sample collection needs to conform to statistics, at least 3 samples are selected, and the histogram lacks standard error.

3. Climatic parameters, such as temperature, humidity, sunshine hours, etc., for traditional and smart greenhouse olive cultivation need to be listed

4. There is a lack of discussion on the differences between this article and previous studies.

5. The article conclusion part lacks the data support.

6.     The article lacks references in the last two years.
